# Influence of Graphene Nanoplatelets on the Performance of Axial Suspension Plasma-Sprayed Hydroxyapatite Coatings

**DOI:** 10.3390/bioengineering10010044

**Published:** 2022-12-29

**Authors:** Pearlin Amaan Khan, Aravind Kumar Thoutam, Vasanth Gopal, Aswin Gurumallesh, Shrikant Joshi, Arunkumar Palaniappan, Nicolaie Markocsan, Geetha Manivasagam

**Affiliations:** 1Centre for Biomaterials, Cellular & Molecular Theranostics (CBCMT), Vellore Institute of Technology, Vellore 632014, India; 2Division of Subtractive and Additive Manufacturing, University West, 461 86 Trollhättan, Sweden; 3School of Advanced Sciences, Vellore Institute of Technology, Vellore 632014, India

**Keywords:** plasma axial suspension plasma spraying, biocompatibility, Ti-6Al-4V, graphene nanoplatelets, hMSCs, hydroxyapatite

## Abstract

Axial suspension plasma spraying (ASPS) is an alternative technique to atmospheric plasma spraying (APS), which uses a suspension of much finer powders (<5-micron particle size) as the feedstock. It can produce more refined microstructures than APS for biomedical implants. This paper highlights the influence of incorporated graphene nanoplatelets (GNPs) on the behavior of ASPS hydroxyapatite (HAp) coatings. The characterization of the ASPS coatings (HAp + varying GNP contents) was carried out using scanning electron microscopy (SEM), energy dispersive spectroscopy (EDS), confocal Raman microscopy (CRM), white light interferometry (WLI), and contact angle measurements. The evaluation of the mechanical properties such as the hardness, roughness, adhesion strength, and porosity was carried out, along with a fretting wear performance. Additionally, the biocompatibility of the Hap + GNP coatings was evaluated using cytotoxicity testing which revealed a decrease in the cell viability from 92.7% to 85.4%, with an increase in the GNP wt.%. The visualization of the cell’s components was carried out using SEM and Laser Scanning Microscopy. Furthermore, the changes in the genetic expression of the various cellular markers were assessed to analyze the epigenetic changes in human mesenchymal stem cells. The gene expression changes suggested that GNPs upregulated the proliferation marker and downregulated the pluripotent markers by a minimum of three folds.

## 1. Introduction

Atmospheric plasma spraying (APS) has been identified as one of the burgeoning orthopedic implant coating techniques alongside electrophoretic deposition (EPD), dip coating, anodization, powder coating, and galvanization [1] mainly due to its validation by the FDA (the Food and Drug Administration) in the USA [2]. APS involves a high-energy plasma arc that can melt the injected powder particles, which will then be propelled with a high velocity onto the target substrate. The deposition onto the substrate compels the particles to solidify into pancake-type splats [3]. Despite its efficacy in the coating, the APS is limited by its phase changes, leading to the generation of by-products such as tri- and tetra calcium phosphates when depositing HAp and producing thicker coatings (>50 µm) [4]. Thinner coatings (<50 µm) with a minimum of 45% of crystallinity is the ISO requisite for implant coatings [5,6]. Suspension plasma spraying (SPS) is a better alternative to APS to achieve the above. Finer ceramic particles with a nanometric to submicrometric size are suspended in a solvent, usually water, ethanol, or a mixture of them, and injected into the plasma plume. Due to the finer particle size feedstock and the inherent mechanisms responsible for the coating formation, SPS can yield coatings with unique microstructures, properties, and a low thickness. SPS can further be categorized based on how the feedstock is injected; a radial injection (feedstock introduced externally and perpendicular to the plasma plume) or axial injection (feedstock introduced axially through the core of the plasma plume). An axial injection for SPS, also known as axial suspension plasma spraying (ASPS) [7], is capable of producing different types of microstructures, such as those which are dense, highly porous, vertically cracked, feathery, etc. [5], and provides a uniform thermal exposure, unlike in the case of a radial injection [7].

Since the inorganic component of the bone tissue comprises hydroxyapatite (HAp) (minimum 60%) [8], HAp coatings on metal implants have been extensively investigated to mimic native apatite. HAp coatings enhance osteoconduction, empower the bioactivity of the bioinert materials, and avert the formation of a connective tissue layer between de novo bone and the implant [4,9,10]. However, due to their intrinsic ceramic properties, these coatings exhibit a poor tensile strength and a low fracture toughness [11,12]. The addition of reinforcement fillers at the nanoscale, such as graphene [13], Al_2_O_3_, and CNTs [14] using the APS technique has been shown to resolve these issues. Among the other reinforced fillers, graphene can drastically influence the mechanical properties even at low stoichiometric amounts [11,15]. Studies show that graphene-based HAp composites have an improved coating crystallinity and exhibit enhanced mechanical properties in terms of the hardness, fracture toughness, and elastic modulus [13]. These influences in the mechanical parameters have motivated this study to employ graphene nano platelets (GNPs)-reinforced hydroxyapatite coating as a medium for an enhanced cell–material interaction.

Despite bestowing bioactive surface properties on inert implants, HAp composite coatings tend to release corrosion debris which can cause an adverse local inflammation in the body [16]. Fretting wear analysis allows for the assessment of the loss of material and the nature of the debris generated during in vitro models to replicate in vivo corrosion behavior. In this study, therefore, a fretting wear study was carried out to evaluate the wear debris generation and the rate of corrosion in fetal bovine serum (FBS).

The bioactivity of the coatings is evaluated by understanding the formation of the bone cell on the coatings. Amongst the various types of cells which are conventionally used to study the efficacy of the coatings, stem cells are considered to be the most appropriate as they are highly coveted for their therapeutic propensity and pluripotency [17]. With more than 2500 completed clinical trials of human mesenchymal stem cells (hMSCs) and more than 1400 ongoing trials (as per clinicaltrials.gov.in), stem cell research remains one of the most thriving researched fields, especially in orthopedic therapeutics. To understand the genetic changes that hMSCs undergo when they come into contact with composite coated implants, we selected hMSCs to assess the in vitro biocompatibility of the ASPS HAp+ GNP coatings on Ti-6Al-4V substrates.

The present work evaluates the coating characteristics, biocompatibility, and genetic changes in the hMSCs cultured on ASPS sprayed HAp + GNP coatings (with a varying GNP content) on a Ti-6Al-4V substrate. The HAp + GNP coatings were characterized using scanning electron microscopy (SEM), Raman spectroscopy, and X-ray diffraction (XRD) and were assessed for their porosity, hardness, roughness, and adhesion strength. A HAp coating with added reinforcers tends to leach into the body fluid, especially during a load-bearing application, leading to the formation of wear debris. A fretting wear analysis was carried out to assess the loss of the material. In addition, the interaction of hMSCs with the coatings was evaluated through cytotoxicity testing and a visualization of the cell adhesion using SEM and laser scanning microscopy, as well as testing for the gene expression using an RT-PCR (reverse transcription polymerase chain reaction).

## 2. Materials and Methods

### 2.1. Preparation of Plasma Sprayed Coatings

An axial suspension plasma spraying technique was employed to deposit hydroxyapatite (HAp)-based coatings on titanium alloy substrates (Ti-6Al-4V). A Mettech Axial III plasma gun and a NanoFeed 350 suspension feeder (Northwest Mettech Corp., Surrey BC, Canada) were the constituents of the ASPS thermal spray system. Disc-shaped titanium alloy (Ti-6Al-4V, AMS-4928W) substrates (BC Metals, Batavia, OH, USA) of a 25.4 mm diameter and 6 mm thickness were used to deposit the HAp-GNP coatings. All the substrate specimens were cleaned with acetone and were grit blasted before thermal spraying using an alumina-based grit media (63 ± 10 μm) at an air pressure of 5.5 bar. This targeted grit blasting process resulted in a surface roughness (R_a_) of approximately 3 µm. The grit-blasted substrates were mounted on a rotating fixture, and the coatings were sprayed to a target thickness of approximately 50 µm. The thermal spray parameters which were employed based on pre-studies to determine suitable spray conditions for the HAp-GNP deposition are summarized in Table 1.

Hydroxyapatite suspensions with and without GNPs were used as feedstock materials in this work. Sample IDs were defined as R1-R4 based on the variation in the suspension constituents, as presented in Table 2. Water-based HAp suspensions were prepared using commercial HAp powders procured from MediPure (Medicoat, France) with a bimodal size distribution (30 vol.% with d50 of 680 μm and 70 vol.% with d50 of 4.7 μm) and GNPs provided by 2D Fab AB, Sweden. The relative amounts of the starting materials were adjusted to yield suspensions containing 20 wt.% HAp with a varying GNP content (0, 0.5, 2, and 5 wt.% of 20 wt.% HAp, respectively), as seen in Table 2. The suspensions were continuously stirred by placing containers on a mechanical roller overnight prior to the spraying to prevent the settling of the suspended particles.

### 2.2. Coating Microstructure

All the samples analyzed in this work were prepared for a microstructure analysis by two-step low viscosity epoxy resin mounting using the vacuum impregnation method. After the first mounting step, the samples were sectioned perpendicular to the coating deposition and second step mounting was performed. The mounted samples were then mirror polished using a semi-automatic Buehler PowerPro 5000 (Buehler, Lake Bluff, IL, USA) machine and gold-sputtered for a cross-sectional SEM analysis. The backscattered electron (BSE) scanning electron microscopy (SEM) and energy dispersive spectroscopy (EDS) analysis were performed on Hitachi TM3000 equipment. The SEM images were then analyzed for their porosity content using grayscale threshold image analysis by ImageJ software. The porosity was determined at low (500×) and high (5000×) magnifications to ascertain both the coarse and fine-scale porosity, respectively. The porosity reported in this work is the average porosity measured over ten images in the case of each sample.

### 2.3. XRD Analysis

The phase constitution of the ASPS Hap coatings on the Ti-6Al-4V surfaces was ascertained using XRD (X’Pert PRO, Malvern PANalytical, Malvern, UK) equipment. The XRD measurements were performed with Cu-Kα (λ = 0.154 nm) radiation, and the 2Ɵ angles were varied from 10° to 90° at an increment of 0.5° at 5 s.

### 2.4. Confocal Raman Analysis

The retention of the GNPs in the HAp coatings was confirmed by the confocal Raman spectrum. An excitation laser wavelength of 532 nm and 9 W power was utilized to obtain the spectrum. A 100× objective lens with a 100 µm slit was used to capture the coating image, and a minimum of five areas were selected for each coating to obtain the spectrum.

### 2.5. Hardness, Adhesion, and Roughness Measurement

A micro indentation technique employing a Vickers hardness (HV) indenter (HMV- Series 2, Shimadzu Corp; Kyoto, Japan) was used to measure the hardness. A load of 245 mN for 10 s was applied to evaluate the HV values of the coatings. The average of the test values is reported from 10 micro indentations. Precautions were taken to ensure that each indentation was at a gap of at least 2.5 times the diagonal length. 

A tensile adhesion test was conducted using a universal MTS Criterion Model 45 (MTS Systems SAS, Creteil, France) tensile testing machine following the EN-582, ASTM-C633 standards. The adhesion samples were placed between the screw heads by applying a thin layer of an adhesive (FM^®^ 1000 epoxy glue, Cytec Industries Inc., Woodland Park, NJ, USA) on the top and bottom counterparts. The glued specimens and the test setup were allowed to cure for about 90 min. The setup was then initiated with a tensile testing machine (MTS Systems SAS, Creteil, France) at a crosshead speed of 1.27 mm/min. Three test samples were evaluated for their adhesion performance to estimate the standard deviation and exactness of the test.

The roughness (Ra) was evaluated by two methods, i.e., a contact-based stylus profilometer and white light interferometry (WLI). The WLI was also used to understand the surface profile of the coating surfaces and was performed using a Profilm 3D instrument (Filmetrics, San Francisco, CA, USA). Contact-based stylus was performed using MITUTOYO SURFTEST-301 profilometer. The EN ISO 4288 standard was followed to report and understand the surface topography and line roughness of the samples. The Ra values analyzed through both equipment are reported as an average of 10 measurement repetitions.

### 2.6. Contact Angle Measurement

The water contact angle (°) of the four coatings R1 (HAp), R2 (HAp + 0.5 wt.%GNP), R3 (HAp + 2 wt.%GNP), and R4 (HAp + 5 wt.%GNP) was measured using the sessile drop method in a contact angle goniometer (HO-IAD-CAM-01A) by Holmarc Opto-Mechatronics, India. The contact angles for all the coatings were noted at time intervals of 10 s, 1 min, and 3 min.

### 2.7. Fretting Corrosion Studies

The fretting corrosion behavior of the coatings was studied using a customized ball on a plate fretting tribocorrosion setup. The customization of the setup was made by integrating the fretting tribometer with the potentiostat (three-electrode cell configuration). In the three-electrode cell system, a standard calomel electrode (SCE), graphite, and coatings/substrate acted as a reference, counter, and the working electrode, respectively. The solution used for the current study is based on the ISO 14242 standard. The solution contains 588 mL of fetal bovine serum (FBS, 30 g/L concentration), 412 mL of phosphate-buffered solution (PBS-ASTM F2129), ethylenediaminetetraacetic acid (EDTA) (0.082 g), TRIS buffer (11.12 g), and sodium azide (0.03%) to retard the bacterial growth. Finally, the solution was adjusted to a pH of 7.4 using diluted HCl. The exposed area of the coating/substrate to the solution was 1 cm^2^, and the remaining area was masked with an acrylic coating. An alumina ball of a 6 mm diameter was used as the counter body. The load, stroke length, frequency, and temperature were fixed at 5 N, ±100 µm, 5 Hz, and 37 ± 2 °C, respectively. All the tests were performed at an open circuit potential (OCP) and repeated thrice to check for the reproducibility.

### 2.8. Biocompatibility Studies

The biocompatibility of the coatings was assessed by checking for the cell viability of the stem cells on top of the coatings, the cytoskeletal imaging using laser scanning microscopy, and a morphological evaluation using SEM. The changes in the expression levels of certain biomarkers were analyzed using RT-PCR to understand the epigenetic influence of the coating. All the samples were rinsed with 70% ethanol prior to the sterilization using steam autoclaving.

#### 2.8.1. Human Stem Cell Culture Expansion and Characterization

Human mesenchymal stem cells (hMSCs) sourced from Wharton’s Jelly were procured from HiMedia (CL001-T25). The HMSCs were expanded in a complete media comprising of minimum essential medium with alpha modification (α-MEM–Gibco Waltham, MA, USA), 10% FBS (Gibco Waltham, MA, USA), and 1% Penstrep (Penicillin/ Streptomycin solution, Gibco) at 37 °C in a humidified atmosphere containing 5% CO_2_. The confluent HMSCs were sub-cultured to the next passage using 0.25% trypsin EDTA (Hi-Clone, GE HEALTH CARE, Marlborough MA, USA) until passage three and were cryopreserved until the next use.

#### 2.8.2. Preparation of Samples for Flow Cytometry

Approximately 100,000 HMSCs were seeded on top of the samples in a six-well plate for a quantitative analysis of the cell viability. After 48 h of incubation, the adhered cells were trypsinized and collected in a micro-centrifuge tube. This was followed by centrifugation at 3000 rpm to remove the supernatant. PBS was added to the tubes, and for every 100 µL of PBS, 1 µL of Propidium Iodide (PI) from Thermofisher, Waltham, MA, USA (P1304MP) was added. After 15 min of incubation at room temperature, centrifugation was again done, and the supernatant with unbound PI was decanted. Lastly, 500 µL of PBS was added to the tube, and the cells were suspended by tapping and before analyzing using Cytoplex Flow cytometry (Beckman Coulter Brea, Brea, CA, USA). BD FACS Diva v. 8.0.1.1 software was used for the analysis, and cells grown in a plastic well without any sample material were used as the control.

#### 2.8.3. Preparation of Samples for SEM

In order to view the cell attachment on the surface of the ASPS samples, the preliminary detection of the cell adhesion and morphology was assessed with a scanning electron microscope (SEM). The HMSCs were counted using a hemocytometer for the SEM analysis and approximately 20,000 cells were seeded on the samples and incubated overnight. Then, the media was aspirated from the samples and the cells were washed with PBS. For the fixation, 1.25% of glutaraldehyde solution was added to the samples and were kept for 30 min of incubation at room temperature. This was followed by a PBS wash and the addition of a 1% OsO_4_ (HiMedia) solution (enough to cover the surface) and incubation for 15 min. Lastly, the samples were washed with PBS twice, and serial dehydration was done using 30%, 50%, 70%, 90%, and 100% ethanol. The samples were dried overnight at room temperature, mounted on aluminum stubs, sputter-coated with gold, and observed under SEM. The SEM investigation was carried out with EVO 15 by Carl Zeiss, Oberkochen, Germany.

#### 2.8.4. Fluorescence Staining Using Phalloidin and DAPI

Similar to the SEM sample preparation, the cells were seeded and cultured for 48 hr followed by washing and fixing with 1.25% glutaraldehyde. After the fixation of the cells on the samples, they were permeabilized using 1% Triton-X 100 for 15 min. Post-permeabilization, the samples were washed twice with PBS and stained with Phalloidin (50 mg/mL) for 40 min, followed by three consecutive PBS washes. For the nuclear staining of the cells, 2 µL of 300 µM of DAPI was added and incubated for 5 min. After the incubation, the cells were observed using an Olympus FluoView FV3000 confocal fluorescence microscope.

#### 2.8.5. Total RNA Isolation and Gene Expression using RT- PCR

The total RNA was extracted after 48 h of incubation using an RNAiso plus Kit (Takara-Clontech, Kusatsu, Shiga, Japan) according to the manufacturer’s protocol and quantified using the BioPhotometer Plus (Eppendorf AG, Hamburg, Germany). A total of 1 μg of RNA was reverse transcribed with a high-capacity cDNA Reverse Transcription Kit (ThermoFisher Scientific, Waltman, MA, USA) using the following thermal program: 25 °C for 10 min, 37 °C for 120 min, 85 °C for 5 min, and put in the hold at 4 °C. To analyze the gene expression of pluripotent markers (NANOG and SOX2), osteogenic markers (RunX2), cytoskeleton (Vimentin, Vinclulin, and Paxillin), and the proliferation marker (Ki67), the primers mentioned in Table 1 were used with SYBR Taq-II (Takara, Japan) as the master mix. The initial denaturation was set at 95 °C for 2 min and 30 s, followed by annealing at 60 °C for 30 s and an elongation at 72 °C for 30 s for 40 cycles. The relative mRNA levels of the gene of interest were normalized to the housekeeping gene, GAPDH, using the ΔΔCt method. All the primers were designed using an oligoanalyzer (Integrated DNA Technologies, Coralville, IA, USA) with default parameters (Integrated DNA Technologies, https://eu.idtdna.com/calc/analyzer) (accessed on 21 November 2020).

#### 2.8.6. Statistical Analysis

All the graphs are presented as the mean ± SD. A statistical analysis was performed using the two tailed student’s *t* test. The level of significance was set at * *p* < 0.05 and ** *p* < 0.001.

## 3. Results

### 3.1. Coating Microstructure and Porosity Content

The backscattered electron (BSE) scanning electron microscopy (SEM) images of samples R1-R4 (see Table 2) are presented in Figure 1, Figure 2, Figure 3 and Figure 4, respectively. All the coatings exhibited a similar coating profile at the interface between the coating and the substrate. Since the deposition parameters are similar, there is no significant variation in the microstructure among the coatings deposited. It is observed that all the coatings are characterized by the presence of undeformed and unmolten/re-solidified particles along with the splats resulting from the impact of the molten particles. The microcracks and pores in the coating system are also evident from the surface morphologies and coating cross-sections, respectively, in each case. Overall, the cross-sectional microstructure of the coatings revealed a refined microstructure along with micro-pores and spherical undeformed particles. Although there are many undeformed particles in the coatings, the particles are well fused to each other. This good cohesion shall be attributed to the bimodal size distribution of the HAp powder used in this study, with the well-molten small particles acting as a binder between the big unmolten particles.

### 3.2. Graphene Retention

The retention of the GNP after axial suspension plasma spraying was initially checked using a high-resolution SEM. The SEM images shown in Figure 5 clearly reveal the presence of GNPs in all the coatings sprayed with a HAp + GNP feedstock (R2, R3 & R4). Further, the HR-SEM images also exhibit an interesting morphology. In the R2 (Hap + 0.5 wt.%GNP) coating, the GNP spans across the crack, which might effectively arrest the crack growth in the coating. In another location in the same coating, the GNP appeared to be protruding from the surface, suggesting that the GNPs could orient themselves in different ways while being entrapped in the coating. In the R3 (HAp + 0.5 wt.%GNP) coating, the GNP was observed in the fractured cross-section. In addition, the folding of the GNP was also observed at the same site. In addition, the EDS elemental mapping was performed on the R3 (HAp + 2 wt.%GNP) coating to confirm the presence of the GNP. Figure 5e clearly reveals that the graphene platelet(s) is embedded in the coating and is surrounded by HAp.

In addition to morphological analysis, a confocal Raman analysis was also performed to study the structure and defects/impurities in the graphene after spraying. Figure 6a shows the Raman spectrum of all the coatings except R1, where HAp is generally Raman inactive in the range of 1200–2600 cm^−1^ due to GNPs being absent in the starting suspension. All the coatings show distinct D, G, and 2D bands of the Graphene. In particular, a prominent G band was observed around 1565 cm^−1^ for all the coatings due to the in-plane vibrational mode involving sp^2^ hybridized carbon atoms that comprise the graphene sheet [18,19]. The D band was also observed to be at around 1332 cm^−1^ for all the coatings. In general, the D band signifies the defect/disorder in the graphene due to the breathing modes of six-atoms rings, which require sprayed coatings defects in order to activate [18,19]. In the present study, the observed D band for all the coatings was weak, suggesting lesser graphene defects despite the plasma spraying. In addition to the D band, a 2D band or second-order D band was also present around 2671 cm^−1^ due to the two-phonon vibration process. This 2D band is always pre sent in the Graphene, and it does not constitute defects.

Further, the I_d_/I_g_ ratio was calculated for all the GNP-incorporated coatings, as shown in Figure 6b. The I_d_/I_g_ ratio values of all the coating were significantly lower than one, which further confirms that the presence of defects in the GNP after spraying was considerably low. Overall, the distinct G, 2D, and a weak D band with low I_d_/I_g_ ratio values suggest the retention of graphene with no significant defect after the axial suspension plasma spraying.

### 3.3. Hardness and Roughness Measurement

The total variation in the porosity content among the different coatings is not significant, as similar spray conditions were used. The porosity content ranges between 20 and 25% in all the samples and is indicated in Figure 6c. It is worth emphasizing that the reported micro indentation-based hardness value are the localized hardness values of the respective coatings. A standard deviation was reported alongside the average hardness from the 10 micro-indentations performed over the samples.

A variation in the hardness values against the respective samples is plotted in Figure 6d. It is observed that samples R2 (HAp + 0.5 wt.%GNP) and R4 (HAp + 5 wt.%GNP) reveal the highest hardness values and are followed by R1 (HAp) and R3 (HAp + 2 wt.%GNP), respectively. The variation in the hardness values between the coatings is insignificant, and the slight differences in the values are complexly related to the feedstock constitution, microstructure, porosity content, etc. The four coatings had a roughness value (Ra) ranging between 4 and 5 microns, consistent with the fact that they were deposited with identical spray parameters, except for the variation in the GNP content in the suspension feedstock.

A comparative plot between the line roughness measured via a contact profilometer and white light interferometer is presented in Figure 6e. It is understood from Figure 6e that the variation between either of the test results is around two microns. This significant variation is attributed to the contact profilometer, where the distinct significance is because the probe could not reach the deep and narrow valleys of the coating deposited. Conversely, the line roughness obtained from 3D WLI images in Figure 7 represents a concentrated small region captured at multiple locations. However, as observed in Figure 6e, the trend seems to follow a similar pattern, which concludes that the line roughness obtained via the contact profilometer and WLI are in a mutual agreement with the closeness of the coating’s roughness. The three-dimensional white light interferometry images of samples R1 (HAp), R2 (HAp + 0.5 wt.%GNP), R3 (HAp + 2 wt.%GNP), and R4 (HAp + 5 wt.%GNP), respectively, are shown in Figure 7. It is observed that the number of peaks and valleys in sample R2 (HAp + 0.5 wt.%GNP) are comparatively higher than the rest. The peaks are highlighted by the red-colored grade on the R2 (HAp + 0.5 wt.%GNP) sample 3D WLI image, whereas such a negligible significance on the surface is not observed among other samples. The color grade is a visual representation of the scale presented next to the images. The other samples showed comparatively smoother areas in the respective 3D WLI images. The presence of a graphene content in the coatings did not influence the surface characteristics significantly.

Figure 8a reveals the actual three adhesion test samples per sample type, respectively, and 8b represents the adhesion strength (MPa) and thickness (µm) plot against samples R1 (HAp), R2 (HAp + 0.5 wt.%GNP), R3 (HAp + 2 wt.%GNP), and R4 (HAp + 5 wt.%GNP), respectively. It is understood that the lowest average adhesion strength is possessed by sample R3 (HAp + 2 wt.%GNP), and later followed by R4 (HAp + 5 wt.%GNP), R2, and R1 (HAp), respectively. The standard deviation reported is from averaging the values of the three samples tested for each type of coating. Earlier studies have shown a significant inverse relation between the thickness and adhesion strength behavior. This adhesion strength data reveals the effective mechanical interlocking between the splats and the substrate asperities obtained via grit blasting. As the feedstock substrate asperities and the spray conditions are the same, the influence of the thickness is a key player in determining the adhesive nature of the deposited coating.

### 3.4. Contact Angle Measurements

The contact angle behavior of the Hap + GNPs coatings was observed by the contact angle measurement. From Figure 8c, it can be clearly inferred that after 3 min, among all the coatings, R3 (HAp + 2 wt.%GNP) exhibits the highest hydrophilicity, and R2 (HAp + 0.5 wt.%GNP) and R4 (HAp + 5 wt.%GNP) exhibit hydrophobicity consistently with a slight decrease. The variation in the contact angle can be attributed to the microstructure, porosity, and roughness values of each sample.

### 3.5. Fretting Corrosion at Open Circuit Potential

Figure 9 shows the fretting corrosion behavior of R1 (HAp), R2 (HAp + 0.5 wt.%GNP), R3 (HAp + 2 wt.%GNP), and R4 (HAp + 5 wt.%GNP) coatings as well as the bare Ti-6Al-4V alloy substrate at the open circuit potential. The potential under the fretting motion is categorized into three regions, i.e., before, during, and after fretting. The potential of the Ti-6Al-4V substrate shows a behavior under the fretting motion that is distinct from that of the coatings. Before fretting, a stable potential was observed due to the formation of a stable oxide layer. At the onset of the fretting, the potentials shifted to a more cathodic potential due to the mechanical removal/destruction of the oxide layer, thereby resulting in the exposure of the bare substrate. During the fretting motion, the potential fluctuates due to the cyclic removal (depassivation) and regrowth (repassivation) of the oxide layer. Finally, as soon as the fretting stopped, the potential increased, suggesting the full recovery of the oxide layer.

On the other hand, the potential of all the coatings remains unaltered throughout the test, which is expected due to the electrochemically inert nature of the HAp. Further, the thickness of the developed coatings is thick enough such that the substrate was not exposed to the solution during the fretting motion. Therefore, from the above results, it can also be inferred that all the coatings are protected under the fretting corrosion condition.

The coefficient of friction (CoF) was monitored along with OCP, shown in Figure 9b. All the coatings possess lower CoF values compared to the substrate. The CoF trend of the coatings exhibits two phases, the run-in wear and steady-state wear. In the run-in wear period, the CoF was found to be higher due to the higher surface roughness (Ra) of the coatings. As time progressed, the surface asperities (i.e., peaks and valleys) in the coating diminished and resulted in a steady-state wear where the CoF was stable. Although all the coatings showed a similar CoF trend, the transition period from the run-in wear to steady-state wear differed. For instance, both R1 (HAp) and R2 (HAp + 0.5 wt.%GNP) displayed a similar transition period (around 2000s). The R3 (HAp + 2 wt.%GNP) coating exhibited a peculiar transition period where the CoF initially decreased, whereas R3 (HAp + 2 wt.%GNP) and R4 (HAp + 5 wt.%GNP) exhibited different transition periods. In particular, the R4 (HAp + 5 wt.%GNP) coating shows the shortest run-in wear period among all the coatings. This different transition period suggests that the concentration of graphene influences the CoF. The worn surface morphology of the Ti-6Al-4V alloy and the coatings after the tribo-corrosion test are shown in Figure 10a–j. A typical wear track (Figure 10a) was observed for the Ti-6Al-4V sample. The worn surface reveals the presence of abrasive grooves along the direction of the fretting. In addition, it also exhibits oxide patches, cracks, and wedges due to plastic flow. On the other hand, the worn surface of all the coatings shows a smooth surface with no signs of grooves. The smooth worn surface suggests that the coatings underwent polishing wear, which is obvious for ceramic coatings despite the soft nature of HAp. Another interesting observation made from the worn surface morphology of the coatings is that the worn scar area (circled) was reduced as the GNP percentage increased. The coating Figure 10i R4 (HAp + 5 wt.%GNP) possesses the lowest wear scare area among all the coatings.

In addition to the morphology of the worn surface, the structural disorder in the GNP after the tribocorrosion test was also analyzed using confocal Raman spectroscopy, shown in Figure 10k, in which the Raman spectrum was recorded at the center of the wear scar and the I_d_/I_g_ ratio was calculated from the spectrum, as shown in Figure 10l. The I_d_/I_g_ ratio of all the coatings was significantly increased after the tribocorrosion test, suggesting that defects were induced in GNPs during these tests. Despite the increased I_d_/I_g_ values, the characteristic Raman spectrum of the GNPs was still evident after the tribocorrosion test, which suggests that the GNP survived even after the simultaneous action of the wear and corrosion.

### 3.6. Cytotoxicity in HAp-GNP Coatings as Determined by Flow Cytometry

The quantitative cell viability of the hMSCs cultured on top of the four ASPS samples was analyzed by PI staining using flow cytometry. The cell viability analysis of the HMSCs is shown in Figure 11 and summarized in Figure 11f. It is inferred that the overall cell number in the ASPS samples decreased when compared to the control. However, the percentage of dead cells within those groups remained lesser than 10% except in the case of R4 (HAp + 5 wt.%GNP), where it was 14.6% when relatively compared to the plastic control and merely 7.6% when relatively compared with the positive control R1 (HAp). The relative percentage of the live cells, when compared with the plastic control for HAp, was 92.7%, HAp + 0.5% GNP was 87.3%, HAp + 2% GNP was 89.6, and HAp + 5%GNP 85.4%. All conditions, respectively do not show any significant cytotoxicity compared to the control (the cells grown in plastic wells).

### 3.7. Adhesion and Morphology of hMSC Grown on the Surface as Determined by SEM and Confocal Microscopy

The adhered HMSCs on the visual examination using an SEM, as shown in Figure 12a–e, looked healthy on the R1 (HAp) (b), R2 (HAp + 0.5 wt.%GNP) (c), R3 (HAp + 2 wt.%GNP) (d), and R4 (HAp + 5 wt.%GNP) (e) surfaces. They were well distributed and exhibited evidence of a typical mesenchymal cell phenotype, making maximum contact with the surface. The fluorescence images Figure 12f–n show the nuclear and cytoskeleton structure in R1 (HAp) (a–c), R2 (HAp + 0.5 wt.%GNP) (d–f), R3 (HAp + 2 wt.%GNP) (g–i), and R4 (HAp + 5 wt.%GNP) (j–l). The blue fluorescence indicates the nucleus of the cell, and the green represents the F-Actin structures of the cells, with the healthy ones retaining the spindle-like shape. Hence, it was concluded that the surface of the ASPS with graphene nanoparticles-sprayed samples provided a healthy environment for the cells to adhere. The surface did not have an acute effect on the HMSCs as they expressed the natural morp hology of large flat cells.

### 3.8. Influence of GNPs on hMSCs Expression Level of Pluripotent Genes

The gene expression of the HMSCs was studied using qPCR. Upon seeding the cells on the surfaces treated with varying ratios of GNP, the pluripotency of the stem cells measured by the NANOG and SOX2 gene was significantly increased by more than five-fold in R4 (Hap + 5 wt.%GNP), as shown in Figure 13a. A significant increase in the stemness of the HMSCs cultured on only the HAp sample (2.51 for Nanog and 1.66 for SOX2) was also observed. For the osteogenic marker RUNX2, a decrease in the fold change of 0.4 was observed for the R2 (HAp + 0.5 wt.%GNP) and R4 (HAp + 5 wt.%GNP) samples. A similar trend of a significant decrease in the fold change of 0.2 for R2 (Hap + 0.5 wt.%GNP) and 0.3 for R4 (Hap + 5 wt.%GNP) in the proliferation marker Ki-67 was also observed. In addition, the assessment of the cytoskeleton markers revealed a significant increase in the expression of Paxillin in the R1 (HAp) sample, while a decrease was witnessed in R2 (HAp + 0.5 wt.%GNP). A significant decrease in the expression level for vimentin and vinculin was observed in all the samples except R3 (HAp + 2 wt.%GNP). Overall, the varying percentage of GNP influences the gene expression of HMSCs.

## 4. Discussion

The present work utilizes axial suspension plasma spraying for depositing hydroxyapatite coatings with and without GNPs on Ti-6Al-4 substrates. The deposited dense coating microstructure is directly attributable to the nature of the feedstock materials and the deposition parameters which are employed. The coating contains undeformed and unmolten particles due to the incomplete melting of the HAp prior to an impact with the substrate. A reason could also be specific to the mechanism of the coating formation in the SPS process. The fine particles in the suspension, while spraying, follow the plasma plume stream and impact the surface of the substrate tangentially. They then grow on the asperities of each other, forming the coating, where the primary dependency is the surface roughness of the underlying substrate roughness (grit blasting was incorporated in this work to maintain the Ra). These particles either sinter together as a large particle or remain undeformed. Another reason might be the bimodal size distribution of the HAp powder used in this study. The small particles are well molten when they impact the substrate acting as a binder for the big particles, which remains unmolten so that a well-fused and fine porosity coating can form. Additionally, the presence in the plasma plume of liquid/water (solvent) reduces the spray temperature and contributes to the less efficient melting of the particles and the consequent presence of undeformed or partially molten particles in the coating [20,21,22,23]. The presence of the GNPs in the coating did not influence the coating deposition or the resulting microstructure because of the very low wt.% addition to the feedstock.

As an illustrative example, EDS mapping on the R3 (HAp + 2 wt.%GNP) coating (Figure 5e) shows the presence of carbon in the clusters, which indirectly confirms the retention of GNPs in the coating after spraying. However, it does not explicitly provide information about the nature of the bonding and defects present in graphene. To study the defects in the graphene that may have manifested during spraying, confocal Raman spectroscopy was performed. The obtained Raman spectrum exhibits a distinct 2D ‘G’ and a weak ‘D’ band for all the coatings, confirming Graphene’s retention after the spraying. Furthermore, the D band that signifies defects was weak for all the coatings, suggesting that fewer defects were present in the graphene after spraying. The plausible reason for the fewer defects could be the very nature of suspension plasma spraying. As already explained, in the axial suspension plasma spraying process, the liquid (water) initially evaporates when exposed to the plasma plume, which lowers the spray temperature and thus consequently prevents the graphene from decomposing inside the plasma plume.

The obtained microstructure, porosity content, hardness, and roughness are depended on the feedstock materials and thermal spray parameters used in this work. The coating deposition parameters and the presence of partially molten particles lead to the high porosity in these coatings (refer to Figure 6c). Similarly, as observed in Figure 6d, the hardness is dependent on the porosity content, where the significant dominance among the samples is not evident, yet the trend followed shows the relation of the hardness to the porosity [5]. For example, the sample R2 (HAp + 0.5 wt.%GNP) has an average low porosity content and higher average hardness. An ideal Hap coating would prefer an increased resistance to the shear surface, enhanced hardness, wear resistance, increased porosity, and a thicker coating for the biomedical applications. The surface roughness plays an important role in defining the adhesion strength as the mechanism of the adhesion of the ASPS coatings depends on the mechanical interlocking of the coatings based on the substrate asperities and the cohesive strength. Here, all the deposition parameters, along with the feedstock and substrate roughness, remain the same; hence the vital factor could be the thickness of the coatings. It is understood that the greater the thickness of the coatings, the greater is the residual stresses, which in the end influences the adhesion properties of the coatings. The variation in the thickness in this work is significant, and the end resulted in the adhesion strength, as observed in Figure 8b. The influence of graphene wt.% did not establish a reasonable correlation between the adhesion strength, but a similar study by Han et al., using electrodeposition instead of plasma spraying established the role of graphene oxide wt.% on the adhesion strength [24]. The graphene oxide/Hap coating on a titanium substrate showed an increasing graphene oxide wt.% in the hydroxyapatite composite coating from a 0 to 12% increased adhesion strength from 6.46 MPa to 17.81 MPa.

In addition to the mechanical properties, the coatings were tested for the corrosion-resistant fretting properties. Under natural conditions, a load-bearing implant is subjected to simultaneous wear and corrosion. In this reference, performing a fretting corrosion study, especially for load-bearing implants like femoral stem, would elucidate the more biologically accurate extrapolation of implant degradation inside the human body. Generally, the passivating metal such as Ti relies on the presence of a stable passive layer for the corrosion resistance properties. However, under mechanical loading, the stable passive layer might damage/remove and expose the bare metal to the corrosive media, resulting in a more metal dissolution. Similarly, in the present study, the sudden shift in the potential of the Ti-6Al-4V alloy toward the cathodic region at the onset of fretting suggests that the passive protective layer was damaged/removed due to the external mechanical loading. Further, during the fretting motion, the potential was found to be fluctuating due to the cyclic depassivation/repassivation, which makes the alloy electrochemically active throughout the fretting cycles.

On the other hand, the ceramic coating is generally electrochemically inert and protects the metal substrate from wear and corrosion. However, defects such as pores and cracks in the coating can lead to a localized corrosion through the capillary mechanism (the ingress of the solution through the pores and the attack of the coating-substrate interface) [25]. In the current study, despite all the coatings possessing pores and cracks, there are no signs of a potential fluctuation throughout the fretting corrosion test, which clearly shows that the substrate was not affected. Further, it was also observed that the presence of GNPs in the coating does not influence the change in the potential during the wear, and it is similar to the R1 coating with no GNPs incorporated. However, the wear scar area was significantly reduced as the GNP percent increased. For instance, the R4 (Hap + 5 wt.%GNP) coating reveals the shallowest wear scar among all the coatings. The reason for such a shallow wear scar is due to the sliding of the GNP layers. It is well known that a weak van der Waals bond between the GNP layers results in sliding when an external force is applied. The GNP layer, which slid during wear, positioned itself adjacent in the direction of the motion and eventually formed a tribofilm, resulting in a lower friction [26,27]. This phenomenon was reflected in the CoF graph, where the GNP-containing coatings show slightly lower CoF values than a pure HAp coating. The above results confirm that the introduction of GNP certainly enhances the tribocorrosion resistance, which is quite important for the longevity of the implants.

Graphene-based composites such as HAp + GNP have a low toxicity [28] and have been shown to promote osteogenesis [12]. The cytotoxicity of different ASPS HAp coatings with GNPs was assessed using flow cytometry. In tandem with other studies on HAp with GNP fillers, no significant cell death was observed for any coating [15]. The relative cell viability for pure HAp exhibited 92% compared to the plastic control, while R4 (HAp + 5 wt.%GNP) with the highest GNP wt.% showed the least percent of the viable cells (85.4%). In an ideal case, a 100% cell viability, as observed in plastic, is expected. However, in reality, some percentage of cells undergo cell death, even when cultured on bioactive and compatible surfaces such as Hap. The realistic relative percentage of Hap+ GNP coatings with respect to only Hap shows approximately 94%, 96%, and 92% for R2 (Hap + 0.5 wt.%GNP), R3 (Hap + 2 wt.%GNP), and R4 (Hap + 5 wt.%GNP), which is much higher when compared to the plastic controls. The adhesion of the mesenchymal stem cells on the axial suspension-sprayed HAp + GNPs is represented by the SEM and fluorescence images in Figure 12, respectively. The characteristic spindle-like morphology is clearly visible in all the fluorescence images, along with well-spread and healthy cells similar to our previous report [3]. Additionally, no apparent morphological changes were observed in the coatings containing GNPs, implying that the varying percentage of GNPs did not contribute to phenotypical changes in the cells. However, the varying percentage of GNPs did influence the genetic changes in the expression of the pluripotency and osteogenic marker.

Studies have shown that titanium substrates coated with graphene highly upregulate the osteogenesis of MSCs compared to native titanium [29]. Similarly, the addition of reduced graphene oxide in scaffolds has shown to improve the cell viability and ALP activity of the stem cells [30,31,32]. A related influence of GNPs was observed in the present study where the cells cultured on the R4 (HAp + 5 wt.%GNP) coating with the highest GNPs content (5 wt.%) exhibited a significant five-fold increase in the expression of NANOG and SOX2, the markers for the stem cell’s pluripotency in comparison to the control. In tandem with the increase in the pluripotent marker, a significant decrease in the osteogenic marker RUNX2 was also observed, implying that the R4 (HAp + 5 wt.%GNP) coating is less suited for orthopedic applications as cells cultured on such a coating tend to favor stemness more than osteogenic differentiation. Interestingly, a significant downregulation of the pluripotent markers was observed in the R2 (HAp + 0.5 wt.%GNP) and R3 (HAp + 2 wt.%GNP) coating when compared to the pure HAp coating without any incorporated GNPs. The above results suggest that the introduction of GNPs in smaller concentrations can down-regulate the pluripotency, but the presence of GNPs in excess R4 (HAp + 5 wt.%GNP) can upregulate. The R3 (HAp + 2 wt.%GNP) coating shows a similar expression of Runx2 as the control and R1 (HAp), but R2 (HAp + 0.5 wt.%GNP) and R4 (HAp + 5 wt.%GNP) show a 0.4-fold decrease. A similar increase in the RUNX2 expression was also observed by Galeh et al., when they incorporated rGO in Zinc-doped Hap inside a polycaprolactone matrix [32]. The cell proliferation marker level of R3 (HAp + 2 wt.%GNP) is significantly higher than in R2 (HAp + 0.5 wt.%GNP) and R4 (HAp + 5 wt.%GNP), implying that R3 (HAp + 2 wt.%GNP) can downregulate the pluripotency while expressing similar and higher levels of Runx2 and Ki67, respectively, when compared to the control and only HAp (R1). With an increase in the GNP content, an increase in the RUNX2 expression is expected; however, in our study, we observed a decline in the expression. Similar observations were previously reported by Jang et al., where 20 µg/mL of graphene oxide and HAp complex showed a significantly lower expression level of RUNX2, OPN, and BSP and for any concentration higher than 10 µg/mL [33]. A trend similar to that of the prozone effect was observed here, where a high concentration of analyte decreased the expression levels due to the saturation of the receptors [34].

## 5. Conclusions

The axial suspension plasma spraying of HAp with a varying wt.% of the GNP (0.5%, 2%, and 5%) was carried out on a Ti-6Al-4V substrate and was evaluated for the mechanical properties of the tribocorrosion behavior and the biocompatibility. The study highlights that all the coatings exhibited comparable mechanical properties and tribological behavior and that no scientifically significant change was observed. However, the in vitro biocompatibility studies showed that the R3 (HAp + 2 wt.%GNP) sample with 2 wt.% of the GNP could upregulate the osteogenic and proliferating markers (RunX2 and Ki67, respectively) and significantly downregulate the pluripotent markers (Nanog and Sox2).

The axial suspension plasma-sprayed HAp with the GNP coating was prepared successfully on a titanium alloy (Ti6Al4V) substrate and evaluated for the fretting wear and biocompatibility with hMSCs.The presence of a varying graphene content in the coatings has no significant influence on the mechanical characteristics. Rather, the obtained microstructure, roughness, porosity, and hardness are correlated to the feedstock material and the respective deposition parameters.The introduction of GNP certainly enhances the tribocorrosion resistance properties by showing a shallower wear scar compared to a pure HAp coating.Altogether, the outcome of the present study indicates that among the varying wt.% of the GNPs, the R3 (HAp + 2 wt.%GNP) coating exhibited an approximately 1.5-fold upregulation of RUNX2 and a 5.7- and 3.5-folds upregulation of the Ki67 gene when compared to R2 (HAp + 0.5 wt.%GNP) and R4 (HAp + 5 wt.%GNP), respectively.Additionally, both R2 (HAp + 0.5 wt.%GNP) and R3 (HAp + 2 wt.%GNP) significantly downregulated by 12.5- and 3-folds of Nanog and 3.8- and 5.5-folds of Sox2 gene expression when compared to R1 (Hap), respectively.

The results of the present study suggest that there could be an optimum GNP concentration that could yield the best biocompatibility for the orthopedic applications, which is determined to be 2 wt.% for the type of GNPs used herein.

## Figures and Tables

**Figure 1 bioengineering-10-00044-f001:**
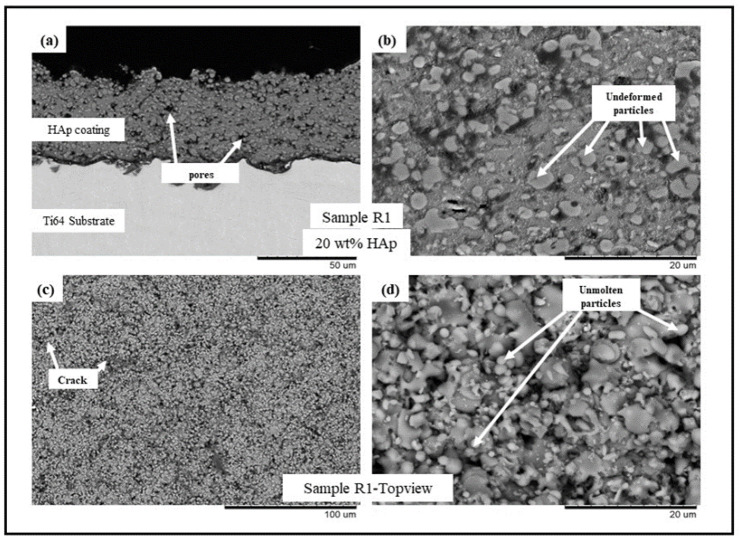
BSE SEM images of sample R1 (HAp) (**a**) cross-sectional view at 1500× magnification, (**b**) cross-sectional view at 5000× magnification, (**c**) top view at 500×, and (**d**) top view at 5000×.

**Figure 2 bioengineering-10-00044-f002:**
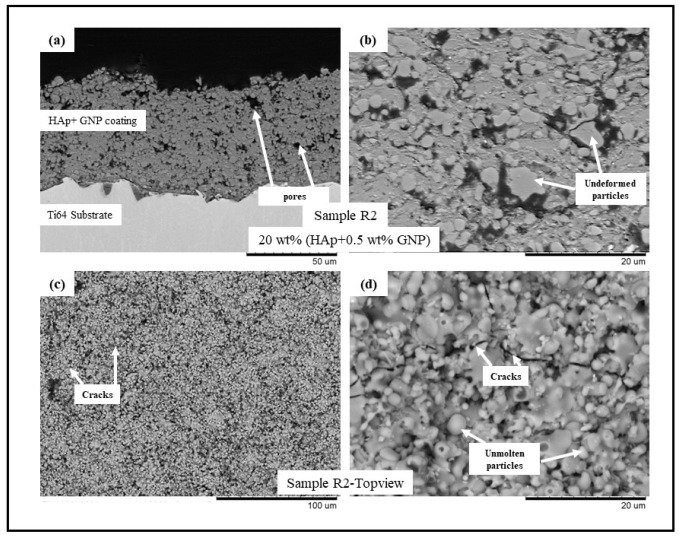
BSE SEM images of sample R2 (HAp + 0.5 wt.%GNP) (**a**) cross-sectional view at 1500× magnification, (**b**) cross-sectional view at 5000× magnification, (**c**) top view at 500×, and (**d**) top view at 5000×.

**Figure 3 bioengineering-10-00044-f003:**
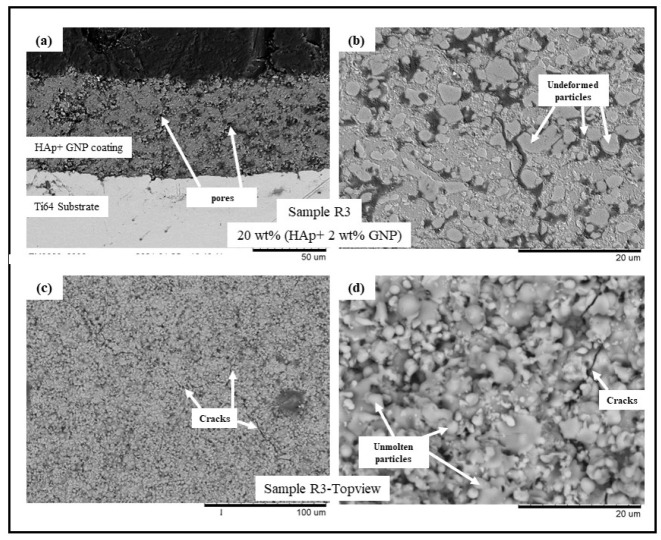
BSE SEM images of sample R3 (HAp + 2 wt.%GNP) (**a**) cross-sectional view at 1500× magnification, (**b**) cross-sectional view at 5000× magnification, (**c**) top view at 500×, and (**d**) top view at 5000×.

**Figure 4 bioengineering-10-00044-f004:**
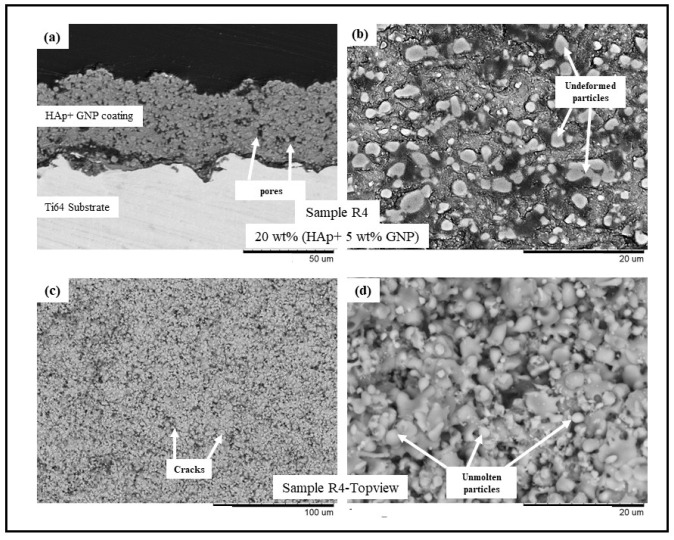
BSE SEM images of sample R4 (HAp + 5 wt.%GNP) (**a**) cross-sectional view at 1500× magnification, (**b**) cross-sectional view at 5000× magnification, (**c**) top view at 500×, and (**d**) top view at 5000×.

**Figure 5 bioengineering-10-00044-f005:**
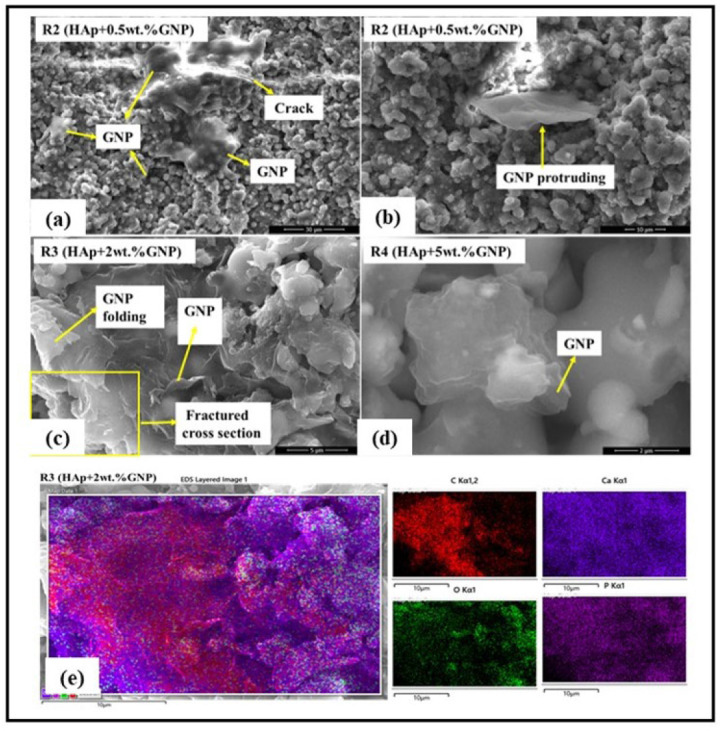
High resolution SEM reveals the presence of GNPs in axial suspension plasma sprayed coatings (**a**–**d**) and EDS elemental mapping on a fractured R3(Hap + 2 wt.%GNP) coating specimen (**e**).

**Figure 6 bioengineering-10-00044-f006:**
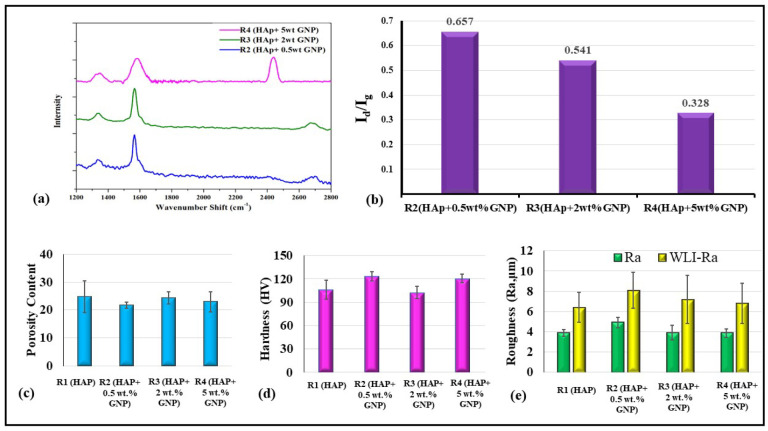
(**a**) Confocal Raman spectra of Axial suspension plasma sprayed samples and (**b**) Id/Ig ratio of R2 (HAp + 0.5 wt.%GNP), R3 (HAp + 2 wt.%GNP) HAp + 2 wt.%GNP and R4 (HAp + 5 wt.%GNP) HAp + 5 wt.%. (**c**) Porosity content (Area%), (**d**) hardness (HV), and (**e**) between line roughness measured via contact profilometer and white light interferometer of samples R1, R2, R3, and R4, respectively.

**Figure 7 bioengineering-10-00044-f007:**
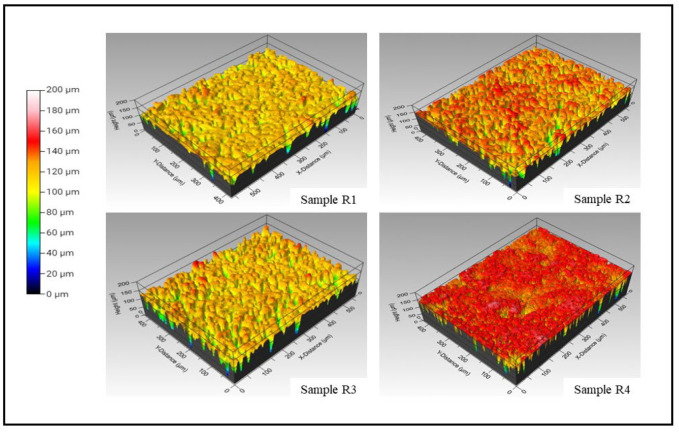
3D white light interferometry images of samples R1 (HAp), R2, R3 (HAp + 2 wt.%GNP), and R4 (HAp + 5 wt.%GNP), respectively.

**Figure 8 bioengineering-10-00044-f008:**
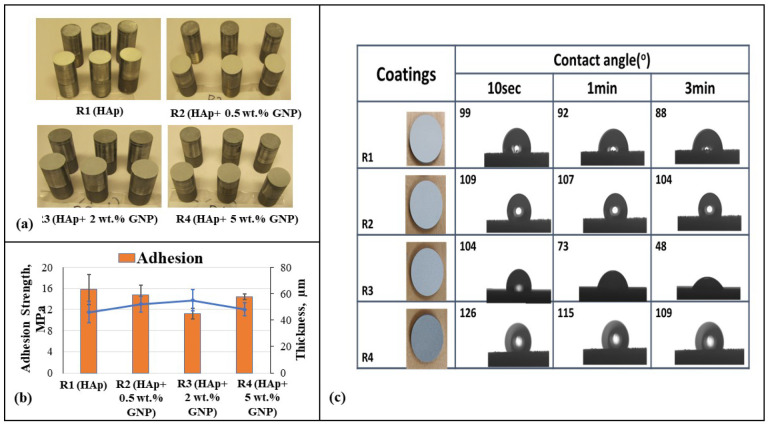
(**a**) Three adhesion test samples per sample type, i.e., R1 (Hap), R2 (Hap + 0.5 wt.%GNP), R3 (Hap + 2 wt.%GNP), and R4 (HAp + 5 wt.%GNP), respectively. (**b**) Adhesion strength and thickness plot against samples R1, R2, R3, and R4, respectively. (**c**) Contact angle measurements with water drop on axial suspension plasma sprayed R1 (HAp), R2 (HAp + 0.5 wt.%GNP), R3 (HAp + 2 wt.%GNP), and R4 (HAp + 5 wt.%GNP) samples after 10 s, 1 min, and 3 min.

**Figure 9 bioengineering-10-00044-f009:**
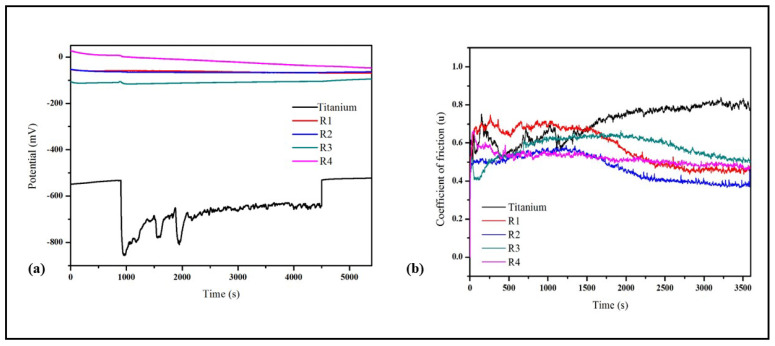
(**a**) Open circuit potential and (**b**) comparison of the friction coefficient of titanium, R1 (Hap), R2 (Hap + 0.5 wt.%GNP), R3 (Hap + 2 wt.%GNP), and R4 (HAp + 5 wt.%GNP).

**Figure 10 bioengineering-10-00044-f010:**
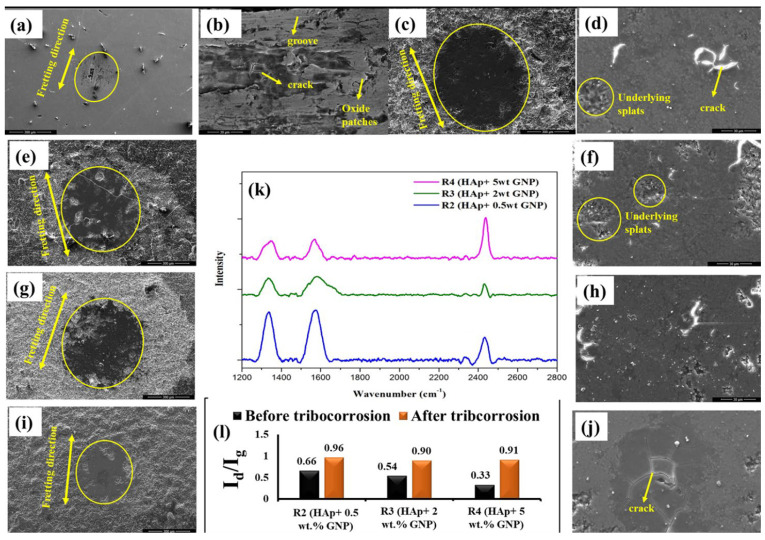
The worn scar morphology of (**a**,**b**) Ti-6Al-4V alloy and the coatings (**c**,**d**) forR1 (HAp), (**e**,**f**) for R2 (HAp + 0.5wt.%GNP), (**g**,**h**) for R3 (Hap + 2 wt.%GNP), and (**i**,**j**) for R4 (HAp + 5 wt.%GNP), after the tribocorrosion test. Graphene retention as assessed by Raman analysis. (**k**) An Id/Ig ratio (**l**) at the site of wear scar post fretting tribocorrosion.

**Figure 11 bioengineering-10-00044-f011:**
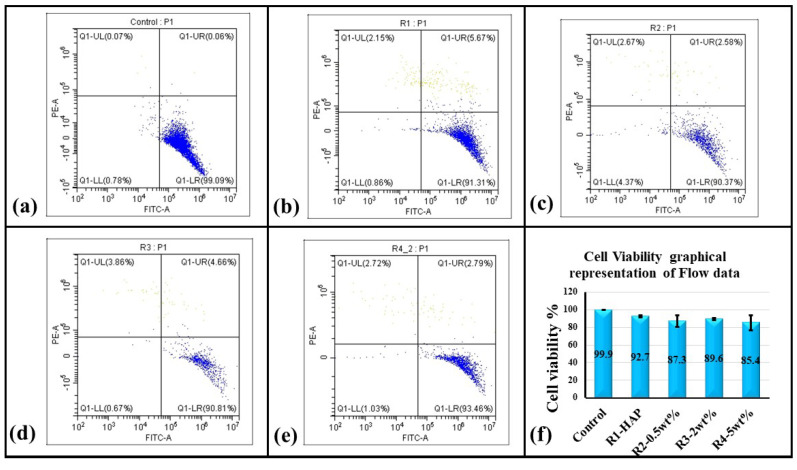
Assessment of cytotoxicity PI staining assay: The viability gate separates the live cells (Blue) from dead cells (grey) in quadrant 4. For (**a**) Control Cells (**b**) R1 (**c**) R2 (**d**) R3 (eR4 (**e**) raphical representation of Flow Cytometry data. Representative (*n* = 4) flow cytometric confirmation of cell viability. (**f**) Graphical representation of cell viability.

**Figure 12 bioengineering-10-00044-f012:**
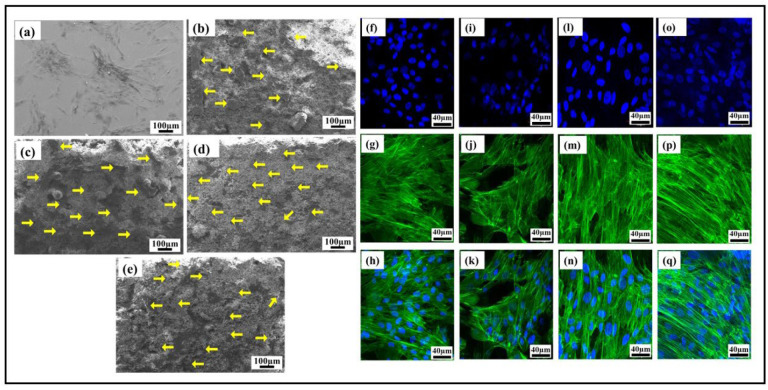
SEM images of hMSCs seeded on samples. The hMSCs cultured on the substrates adhered to plastic control (**a**) and the surface of sample R1 (HAp) (**b**), R2 (HAp + 0.5 wt.%GNP) (**c**), R3 (HAp + 2 wt.%GNP) (**d**), and R4 (HAp + 5 wt.%GNP) (**e**). Yellow arrows indicate cells. Confocal images of hMSCs cultured on samples. Nuclear and F-actin staining of hMSCs using DAPI in blue color and Phalloidin in green color for cells seeded on R1 (HAp) (**f**–**h**), R2 (HAp + 0.5 wt.%GNP) (**i**–**k**), R3 (HAp + 2 wt.%GNP) (**l**–**n**), and R4 (HAp + 5 wt.%GNP) (**o**–**q**).

**Figure 13 bioengineering-10-00044-f013:**
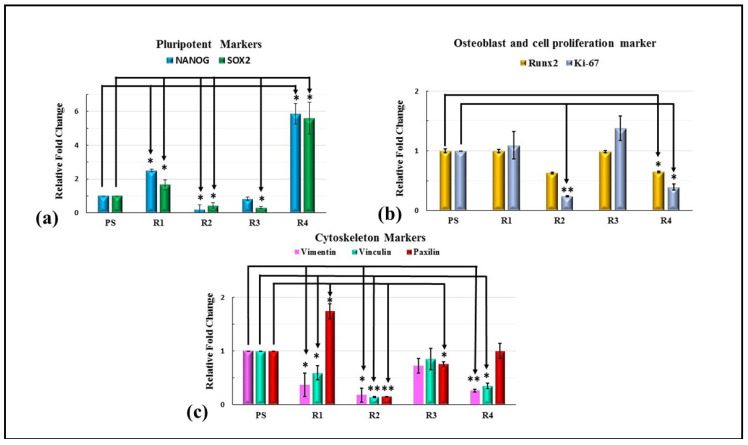
Quantitative evaluation of expression of pluripotent, osteogenic, proliferation, and cytoskeleton markers. Relative fold change (normalized to 1 for control) for (**a**) pluripotent markers (Nanog and SOX2), (**b**) osteogenic markers (Runx2), and proliferation marker (Ki-67) and (**c**) cytoskeleton markers (vinclulin, vimentin, and paxilin). Values expressed are the mean ± SD of 3 independent runs. Where * *p* < 0.05 and ** *p* < 0.001.

**Table 1 bioengineering-10-00044-t001:** Axial suspension plasma spray deposition parameters.

Spray Parameter	Value
Power, kW	75
Current, A	220
Spray distance, mm	70
Suspension feed rate, mL/min	40
Target coating thickness, µm	50
Nozzle, inch	5/16

**Table 2 bioengineering-10-00044-t002:** Sample ID, suspension constituents and coating thickness of samples.

Sample ID	Suspension Constituents	Coating Thickness, µm
R1	20 wt.% HAp + Distilled Water	46 ± 8
R2	20 wt.% (Hap + 0.5 wt.%GNP) + Distilled Water	52 ± 6
R3	20 wt.% (Hap + 2 wt.%GNP) + Distilled Water	55 ± 8
R4	20 wt.% (Hap + 5 wt.%GNP) + Distilled Water	48 ± 5

## Data Availability

No new data were created or analyzed in this study. Data sharing is not applicable to this article.

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
