# Peer review of "Influence of Graphene Nanoplatelets on the Performance of Axial Suspension Plasma-Sprayed Hydroxyapatite Coatings"

_bioengineering, 2022, doi:10.3390/bioengineering10010044_

Round 1
Reviewer 1 Report
Only fewer mistakes need to be corrected:
1. The acronyms should be given their full name when they first appear only. such as, HAp, RT-PCR, EDTA,etc. Please check the whole text.
2. In table 1, the unit of nozzle parameter is missing.
3. what the temperature of the thermal spray?
4. The scaleplate in fig 7 is not clear.
5. Scheme 8? It should be fig.8?
6. Is there any different for the contact angle measurement use water or body fluid?
7. The caption of Fig12, (e). Pink arrows indicate cells, well, the in the picture, it is yellow arrows.
Author Response
Reviewer #1:
Only fewer mistakes need to be corrected:
- The acronyms should be given their full name when they first appear only. such as, HAp, RT-PCR, EDTA,etc. Please check the whole text.
Reply:- We agree with reviewers comment and have made changes accordingly. The acronyms are expanded when they appear first. ( Page 1, Line 17; Page 2 Line 91; Page 4 Line 177)
- In table 1, the unit of nozzle parameter is missing.
Reply:- We thank the reviewer for their comment The unit is: inch
- what the temperature of the thermal spray?
Reply:- We have not done temperature measurements on the spray plume, particularly for these spray trials. However, as shown in [20] the spraying temperature can significantly drop in the SPS (Suspension Plasma Spray) process as compared to APS (Atmospheric Plasma Spraying). The spray temperature in SPS can be as low as 2100 degC [20] whereas in APS the spraying temperatures are >3800 degC.
Obs.: When SPS data/results are shown/published they usually are compared to APS, as SPS is a derivate of APS, the main difference being the feedstock. In SPS the feedstock is a suspension (fine HAp powder in water, in this case) while in APS the feedstock is dry powder (with higher size).
- The scale plate in fig 7 is not clear.
Reply:- We thank the reviewer for the pointing this out. We have reuploaded the figure separately with better quality.
- Scheme 8? It should be fig.8?
Reply:- Thank you for pointing it out. The error has been rectified. (Page11, Line 359)
- Is there any different for the contact angle measurement use water or body fluid?
Reply:- We believe , that definitely there will be a difference in the contact angle when we use water or body fluid. For example, we used PBS+BSA (albumin protein) as a body fluid in the current study. The introduction of albumin protein changes the viscosity of the solution, which directly influences wettability. Also, the albumin protein is a negatively charged protein that controls surface energy. Since there are various factors involved that directly affect wettability when using body fluid, we have performed the contact angle measurement in distilled water for simplicity. However, we strongly acknowledge that for in vivo conditions, a relevant body fluid solution will be ideal for conducting contact angle measurements.
- The caption of Fig12, (e). Pink arrows indicate cells, well, the in the picture, it is yellow arrows.
Reply:- Thank you for pointing it out. The error has been rectified. (Page16, Line 474)
Reviewer 2 Report
This manuscript provide a systematic presentation on sprayed HA coating. However, for HA coatings application, a big and key problem is their adhesive and cohesive strength. In this work, a improved strength by addition of Graphene is not shown. At the same time, the wetting angles of the as-reveived coatings are higher than that of commonly reported, can author give the reason?
Author Response
This manuscript provides a systematic presentation on sprayed HA coating. However, for HA coatings application, a big and key problem is their adhesive and cohesive strength. In this work, a improved strength by addition of Graphene is not shown. At the same time, the wetting angles of the as-received coatings are higher than that of commonly reported, can author give the reason?
Reply:- This study mostly focused on assessing the capability of the SPS process to produce HA coatings whit incorporated graphene nano-pellets (GNP), given the difficulty to preserve graphene in coatings produced by high temperature processes. In this respect, no further process optimization was carried out to increase the adhesion/cohesion of the coatings. However, in a previous study done by the authors [5], HA coatings produced by ASPS i.e. in similar conditions as in this study, revealed adhesions of 42 ± 10 MPa [5].
Regarding the second part of the question on Contact Angle (CA), we hypothesize that the higher CA could be attributed to the bimodal size distribution of the powder used in suspension i.e. 30 vol.% with d50 of 680 μm and 70 vol.% with d50 of 4.7 μm. With this powder, the resulting roughness (surface profile) of the coatings was a combination of both nano- and micrometric features that could have directly affected the hydrophobicity of the coatings. However, more research is needed to confirm this assumption.
Reviewer 3 Report
The manuscript titled Influence of Graphene nanoplatelets on the behavior of axial suspension plasma-sprayed (ASPS) hydroxyapatite coatings in view of biomedical applications. In general, ASPS is an alternative technique to atmospheric plasma spraying (APS), which uses a suspension of much finer powders (< 5-micron particle size) as feedstock that produces more refined microstructures than APS for biomedical implants. The authors have investigated the performance of the influence of incorporated graphene oxide nanoplatelets (GNPs) on the behavior of ASPS hydroxyapatite (HAp) coatings. The authors have varied GNP content and characterized the ASPS coatings (HAp + varying GNP contents) that have been carried out using SEM, EDS, Raman microspectroscopy, optical Interferometry, and contact angle measurements. The authors also investigated the performance and evaluated mechanical properties including hardness, roughness, adhesion strength, and porosity along with fretting wear performance. Additionally, the biocompatibility of HAp + GNP coatings was evaluated using cytotoxicity testing and visualization of cell components using SEM and Laser scanning microscopy. Finally, the changes in the genetic expressions of various cellular markers were assessed to analyze the epigenetic changes in human mesenchymal stem cells.
Despite some interesting findings that authors reported for biomedical implants, there are some demerits that require the authors' attention. First and foremost, the quality of all the figures is very low, the legends are too small to be read. Then there are too many figures, therefore some of the figures need to be consolidated and some can be put in as supplementary materials. The authors need to re-read the entire manuscript for taking care of mechanical deficiencies. Once these changes are made, the paper can be considered for publication.
Author Response
Reviewer #3:
The manuscript titled Influence of Graphene nanoplatelets on the behavior of axial suspension plasma-sprayed (ASPS) hydroxyapatite coatings in view of biomedical applications. In general, ASPS is an alternative technique to atmospheric plasma spraying (APS), which uses a suspension of much finer powders (< 5-micron particle size) as feedstock that produces more refined microstructures than APS for biomedical implants. The authors have investigated the performance of the influence of incorporated graphene oxide nanoplatelets (GNPs) on the behavior of ASPS hydroxyapatite (HAp) coatings. The authors have varied GNP content and characterized the ASPS coatings (HAp + varying GNP contents) that have been carried out using SEM, EDS, Raman microspectroscopy, optical Interferometry, and contact angle measurements. The authors also investigated the performance and evaluated mechanical properties including hardness, roughness, adhesion strength, and porosity along with fretting wear performance. Additionally, the biocompatibility of HAp + GNP coatings was evaluated using cytotoxicity testing and visualization of cell components using SEM and Laser scanning microscopy. Finally, the changes in the genetic expressions of various cellular markers were assessed to analyze the epigenetic changes in human mesenchymal stem cells.
Despite some interesting findings that authors reported for biomedical implants, there are some demerits that require the authors' attention. First and foremost, the quality of all the figures is very low, the legends are too small to be read. Then there are too many figures, therefore some of the figures need to be consolidated and some can be put in as supplementary materials. The authors need to re-read the entire manuscript for taking care of mechanical deficiencies. Once these changes are made, the paper can be considered for publication.
Reply:- We appreciate the constructive suggestion. Due to the constraint in the template legend all images were made to fit and thus were difficult to read. However, in the revised version images have been enlarged and uploaded with higher resolution. We agree with the reviewer that some figures can be consolidated for example Fig 1-4 the SEM images of all 4 coatings. However, consolidating them would again miniaturize them making it difficult to read and observe.
As per reviewer’s suggestion the manuscript has been rectified for any grammatical – typo error. In addition to the above comments, all spelling and grammatical errors have been corrected.
Reviewer 4 Report
From my perspective, this is an interesting work; however, the following issues have to be addressed carefully.
- In the abstract, you need to focus more on quantitative information, not qualitative ones.
In the introduction section:
- Which new achievement(s) was supposed to be obtained by the present research compared to the previous reports?
- The authors should extensively investigate other coating methods compared to the plasma spray and state their advantages and disadvantages. The following references can be used for studying:
https://doi.org/10.1016/j.surfcoat.2019.02.029
https://doi.org/10.1016/j.ceramint.2022.11.135
https://doi.org/10.1016/j.surfcoat.2020.125858
In the Results and discussion section:
· In section 3.8. "Influence of GNPs on hMSCs expression level of pluripotent genes", Authors strongly suggested improving the discussion. To enhance the discussion about the effect of hydroxyapatite and graphene on osteogenesis, you can study the following references:
https://doi.org/10.1016/j.cej.2021.131321
https://doi.org/10.1016/j.msec.2020.111102
https://doi.org/10.1016/j.bioactmat.2020.10.017
Author Response
From my perspective, this is an interesting work; however, the following issues have to be addressed carefully.
- In the abstract, you need to focus more on quantitative information, not qualitative ones.
Reply:- We appreciate reviewers the constructive suggestion and have re-edited the abstract to reflect more quantitative information than qualitative. (Page 1 Line 22-23, Line 26-27)
In the introduction section:
- Which new achievement(s) was supposed to be obtained by the present research compared to the previous reports?
Reply:- As reflected by the points in the conclusion section, the present research suggest that there could be an optimum GNP concentration that could yield the best biocompatibility for orthopedic applications, which is determined to be 2 wt.% for the type of GNPs used herein.
- The authors should extensively investigate other coating methods compared to the plasma spray and state their advantages and disadvantages. The following references can be used for studying:
https://doi.org/10.1016/j.surfcoat.2019.02.029
https://doi.org/10.1016/j.ceramint.2022.11.135
https://doi.org/10.1016/j.surfcoat.2020.125858
In the Results and discussion section:
- In section 3.8. "Influence of GNPs on hMSCs expression level of pluripotent genes", Authors strongly suggested improving the discussion. To enhance the discussion about the effect of hydroxyapatite and graphene on osteogenesis, you can study the following references:
https://doi.org/10.1016/j.cej.2021.131321
https://doi.org/10.1016/j.msec.2020.111102
https://doi.org/10.1016/j.bioactmat.2020.10.017
Reply:- We thank the reviewer for the constructive suggestion. To enhance the discussion, the relevant above mentioned articles were studied and are cited in the modified discussion Page 18 Line 537-542; Page 19, Line 595-597, 610-612).